# The Role of Beta Cell Recovery in Type 2 Diabetes Remission

**DOI:** 10.3390/ijms23137435

**Published:** 2022-07-04

**Authors:** Mara Suleiman, Lorella Marselli, Miriam Cnop, Decio L. Eizirik, Carmela De Luca, Francesca R. Femia, Marta Tesi, Silvia Del Guerra, Piero Marchetti

**Affiliations:** 1Department of Clinical and Experimental Medicine, University of Pisa, 56126 Pisa, Italy; mara.suleiman@for.unipi.it (M.S.); lorella.marselli@unipi.it (L.M.); carmela.deluca3288@gmail.com (C.D.L.); marta.tesi91@gmail.com (M.T.); silvia.delguerra@unipi.it (S.D.G.); 2ULB Center for Diabetes Research, Université Libre de Bruxelles, 1050 Brussels, Belgium; cnop@ulb.ac.be (M.C.); deizirik@ulb.ac.be (D.L.E.); 3Division of Endocrinology, ULB Erasmus Hospital, Université Libre de Bruxelles, 1050 Brussels, Belgium; 4Departmental Section of Endocrinology and Metabolism of Transplantation, AOUP Cisanello Hospital, 56124 Pisa, Italy; frfemia@gmail.com

**Keywords:** type 2 diabetes, lipotoxicity, glucotoxicity, remission, pancreatic islets, pancreatic beta cells, insulin secretion, low-calorie diets, bariatric surgery, transcriptome

## Abstract

Type 2 diabetes (T2D) has been considered a relentlessly worsening disease, due to the progressive deterioration of the pancreatic beta cell functional mass. Recent evidence indicates, however, that remission of T2D may occur in variable proportions of patients after specific treatments that are associated with recovery of beta cell function. Here we review the available information on the recovery of beta cells in (a) non-diabetic individuals previously exposed to metabolic stress; (b) T2D patients following low-calorie diets, pharmacological therapies or bariatric surgery; (c) human islets isolated from non-diabetic organ donors that recover from “lipo-glucotoxic” conditions; and (d) human islets isolated from T2D organ donors and exposed to specific treatments. The improvement of insulin secretion reported by these studies and the associated molecular traits unveil the possibility to promote T2D remission by directly targeting pancreatic beta cells.

## 1. Introduction

Pancreatic beta cell failure, due to the interplay of genetic and acquired factors, is key to the onset and progression of type 2 diabetes (T2D) [1,2,3,4,5] (Figure 1). Beta cells are unique endocrine cells that synthesize, store and secrete insulin under the control of multiple integrated signals, thereby tightly regulating blood glucose concentrations. They have an average diameter of 10 μm and contain approximately 20 pg insulin. Beta cells are the most common cell type in the human pancreatic islets, comprising 50–80% of islet cells. Studies with autopsy samples, organ donor specimens and surgical cases have found that the human beta cell mass may vary from 0.6 to 2.1 g, with the beta cell volume and area (relative to the pancreatic tissue) ranging from 1.1 to 2.6% and 0.6 to 1.6%, respectively. In healthy individuals, beta cells release about 30–70 units of insulin per day (essentially depending on body weight, physical activity and nutritional habits), half of which under basal conditions and half in response to meals. The most important regulator of insulin release is glucose, acting as both a trigger and an amplifier of insulin secretion. Several other molecules regulate insulin secretion, including non-carbohydrate nutrients, hormones and neurotransmitters.

In T2D, representing 80–90% of all cases, beta cell incompetence is due to both reduced mass and functional impairment [3,4,5,6,7,8,9,10,11]. Studies ex vivo and in vitro show that dysfunction, rather than death, is the prevalent beta cell defect in T2D [10,11,12]. Hence, one could expect that under certain conditions the impaired beta cell insulin secretion might recover. A recent consensus reports the definition and interpretation of “remission” in T2D patients. The authors propose HbA1c < 48 mmol/mol (6.5%) measured at least 3 months after cessation of glucose-lowering pharmacotherapy as the diagnostic criterion for remission [13,14] (Table 1). This improvement in glucose levels back to a normal or near-normal interval can be achieved by lifestyle changes or following a number of interventions [13,14]. A recent observational study in 162,316 T2D subjects reports that remission occurred in 5% of cases, and that factors associated with remission were older age, HbA1c < 48 mmol/mol (6.5%) at diabetes diagnosis, no previous history of glucose-lowering therapy, weight loss and bariatric surgery [15].

Here, we review the in vivo and ex vivo evidence and discuss the perspectives of remission of T2D, focusing on the role of beta cells.

## 2. Reversibility of Beta Cell Functional Damage: In Vivo Evidence in Non-Diabetic Subjects

Since the UK Prospective Diabetes Study [16], it has been assumed that the decline in beta cell functional mass begins before the onset of T2D and proceeds relentlessly thereafter, leading to worsening glycemic control and requiring progressive intensification of diabetes therapy, often culminating in the need for exogenous insulin therapy. The causes of this deterioration are not completely understood, but prolonged exposure to saturated fatty acids (lipotoxicity), high glucose (glucotoxicity) or combinations thereof (lipoglucotoxicity) in genetically predisposed individuals have been proposed to contribute to beta cell failure, probably via mitochondrial dysfunction, endoplasmic reticulum stress, oxidative stress, loss of cell identity and other mechanisms [17,18,19,20,21,22,23,24,25,26,27,28] (Figure 1).

Growing evidence shows that alleviation of metabolic stress can improve beta cell function and even induce remission of T2D (Table 2). In the 1990s, a study investigated the short- and long-term effects of lipid infusion on insulin secretion [29]. Twelve healthy individuals underwent a 24 h Intralipid (10% triglyceride emulsion) infusion. After an overnight fast (baseline), at 6 and 24 h of intravenous Intralipid administration and 24 h after Intralipid discontinuation (recovery test), they underwent an intravenous glucose tolerance test. The Intralipid infusion increased by threefold the plasma non-esterified fatty acid (NEFA) concentrations, with no difference between the 6 and 24 h timepoints. Compared to the baseline, the acute insulin response to glucose was increased at 6 h but it decreased at 24 h. After the 24 h recovery phase, fasting plasma NEFA concentrations and the acute insulin response to glucose had returned to baseline values, demonstrating the reversibility of beta cell functional alterations induced by in vivo “lipotoxicity”.

It was later shown that lipotoxicity particularly impacts subjects with familial predisposition to diabetes [36]. Insulin secretion was evaluated during a 4-day lipid infusion in normal glucose-tolerant individuals with or without a family history of T2D. Thirteen and eight subjects, respectively, received in random order a lipid (Liposyn III, 20% triglyceride emulsion) or saline infusion. On Days 1 and 2, the insulin and C-peptide levels were measured after standardized mixed meals, and a hyperglycemic clamp was performed on Day 3. Similar concentrations of NEFA were observed in the two groups. In subjects without a family history, the lipid infusion significantly increased insulin secretion after mixed meals and during the hyperglycemic clamp. On the contrary, in the group with a T2D family history, both first- and second-phase insulin release decreased markedly. These alterations became even more pronounced after correction for insulin sensitivity. Subjects who are genetically at risk to develop T2D are hence susceptible to beta cell functional damage induced by increased plasma fatty acids, resulting in reduced insulin secretion in response to a mixed meal and intravenous glucose challenge. In a follow-up study, the same group demonstrated improved beta cell function after NEFA lowering with acipimox, an antilipolytic nicotinic acid derivative [30]. Nine non-diabetic participants with a strong predisposition to T2D received in random order the drug or placebo for two days in a double-blind crossover design. Acipimox reduced the plasma NEFA levels by a third after 48 h, which was associated with improved beta cell function during mixed meal tests. First and, more evidently, second-phase insulin secretion during the hyperglycemic clamp also improved, which was further enhanced after adjustment for the prevailing insulin resistance. This study provides further evidence that lipotoxicity can impair beta cell function, at least in individuals predisposed to T2D, and that beta cell functional damage can recover, provided the metabolic insult is attenuated.

## 3. Reversibility of Beta Cell Functional Damage: In Vivo Evidence in T2D Subjects

Whether and how recovery of beta cell function may also apply to patients with overt T2D and impact the clinical trajectory has been addressed in a few studies (Table 2). Remission of diabetes of variable duration can be achieved in T2D subjects by carbohydrate restriction and low-calorie diets (usually associated with physical exercise), pharmacological treatments and bariatric surgery [37,38,39,40,41,42]. Improved beta cell function has been described in most studies documenting diabetes remission. In early work [31], 11 T2D subjects were examined before and after 1, 4 and 8 weeks of a low-calorie (600 kcal/day) diet. After 1 week of restricted energy intake the fasting plasma glucose normalized. First-phase insulin secretion increased during the study and approached the control values. Maximal insulin response became supranormal at 8 weeks. In parallel, the pancreatic triacylglycerol content decreased. In the more recent Diabetes Remission Clinical Trial (Direct), T2D remission and persistence of non-diabetic blood glucose control were achieved in 46% of patients on a low-calorie diet [37]. In a sub-study of 64 and 26 people from the intervention and control groups, respectively [43], the pancreatic fat content decreased whether glycemia normalized or not. Recovery of first-phase insulin release was seen in individuals with diabetes remission, which was durable at one year. Interestingly, subjects with sustained diabetes remission compared to those with relapse had less hepatic VLDL1-triglyceride production and VLDL1-palmitic acid content, no re-accumulation of pancreatic fat and maintained the first-phase insulin response by 2 years [44]. Persistence of remission was associated with improved pancreas morphology and declining pancreatic fat [45]. Sex does not seem to affect changes in intrapancreatic fat and VLDL1-triglyceride production after weight loss [46].

The possible beneficial effects of glucose-lowering drugs [47,48] on beta cells of T2D patients have been extensively reviewed recently [9,49,50] and are not addressed in detail here. Metformin, pioglitazone, DPP-4 inhibitors, GLP-1 receptor agonists and exogenous insulin, alone or in combination, have been found to enhance beta cell function. Insulin release increases by 40–70% in individuals at high risk of T2D or with newly diagnosed diabetes [50]. The mechanisms are unclear, but beta cell function may improve without major changes in insulin sensitivity [51,52] or glycemic control [53]. With rare exceptions, the beneficial actions of most of these treatments disappear shortly after drug discontinuation [32,54], indicating limited if any disease-modifying effects.

In the case of bariatric surgery, Roux-en-Y gastric bypass, vertical sleeve gastrectomy and biliopancreatic diversion improve glucose control and can promote remission of T2D [33,55,56,57,58,59,60], which is more marked in the presence of residual beta cell function [57]. This may occur independently of weight reduction [55,56], and the beneficial effects on glycemic indices may persist over years and become more evident as weight loss progresses. Bariatric surgery has pleiotropic effects on organs, and in subjects with T2D rapid improvement in insulin secretion is seen, which persists over time in many but not all patients [34,61,62]. Restoration of first-phase insulin release to intravenous glucose administration (1–4 weeks after Roux-en-Y gastric bypass or biliopancreatic diversion) has been consistently observed, may occur before any significant weight loss and contributes to diabetes improvement or remission [35,63,64]. Although the mechanisms promoting beta cell function after bariatric surgery are still largely unclear, increased GLP-1 release and potentiated incretin effect are believed to play a major role [65,66].

Altogether, these in vivo results demonstrate that beta cell function can recover in some T2D subjects, to achieve and possibly sustain remission of diabetes.

## 4. Reversibility of Beta Cell Functional Damage: Ex Vivo Evidence with Non-Diabetic Islets

In the past few years, the reversibility of beta cell damage has been directly tested in isolated human islets studied ex vivo (Table 3). The use of human islets, prepared from the pancreas of organ donors, allows to assess beta cell features independently from confounding in vivo factors, and, importantly, shed light on islet cell morphological and molecular traits.

In an early study [67], islets from 7 non-diabetic donors were cultured for 48 h in normal (5.5 mmol/L) or high (16.7 mmol/L) glucose-containing medium. Islets were perifused with 3.3 and 16.7 mmol/L glucose or 10 mmol/L L-arginine. The islets cultured at high glucose lost acute glucose-stimulated insulin release but preserved the response to arginine, supporting the concept of a selective loss of beta cell sensitivity to glucose induced by glucotoxicity. Notably, after 48 additional hours of culture in medium with 5.5 mmol/L glucose, the islets partially recovered insulin secretory function. A few years later, these findings were confirmed in a study in which non-diabetic donor islets were exposed to 33 mmol/L glucose for 4 and 9 days [68]. After this glucotoxic culture, a severe decrease was seen in insulin content (related to reduced insulin mRNA and PDX1transcriptional activity) and glucose-stimulated insulin release. Interestingly, most of these beta cell alterations were partially reversible when islets previously cultured in high glucose were transferred to 5.5 mmol/L glucose for 3 days.

Recently, a comprehensive study was performed with a large number of human islet preparations to assess the direct impact of lipoglucotoxic treatments on beta cell function and to evaluate if the effects were persistent or reversible after washout [28]. The ex vivo lipoglucotoxic treatments were selected to reflect pathophysiologically relevant concentrations of the most common saturated fatty acid palmitate and glucose [72]. After the isolation, the islets were kept in control medium (containing 5.5 mmol/L glucose) for 2 days. They were then cultured for 2 days in the presence of the metabolic stressors 11.1 mmol/L or 22.2 mmol/L glucose and 0.5 mmol/L palmitate, alone or in combination. Subsequently, the stressor was washed out and the islets were cultured in normal medium for 4 more days. Glucose-stimulated insulin secretion declined after incubation with palmitate and/or 22.2 mmol/L glucose. Of interest, after the washout, recovery of insulin secretory function was observed for palmitate or high glucose alone, but not for the combination of the two, indicating that beta cell dysfunction induced by metabolic stress is reversible under certain conditions and sustained in others. Transcriptome analysis of islets exposed to palmitate and/or high glucose identified several hundred differentially expressed genes involved in metabolic pathways, endoplasmic reticulum stress and inflammation. Interestingly, the gene expression signatures induced by metabolic stressors and/or washout overlapped, at least in part, with transcriptomic patterns of T2D islets [28].

These results corroborate directly at the islet cell level the in vivo findings in humans, and, importantly, shed light on the potential underlying mechanisms. More prolonged exposure (a few weeks) of human islets to a hyperglycemic environment (such as after transplantation into diabetic immune deficient mice) may be associated with more profound beta cell functional changes [69].

## 5. Reversibility of Beta Cell Functional Damage: Ex-Vivo Evidence with T2D Islets

A few studies have assessed whether the defects of islets from T2D donors can be counteracted (Table 3). The first data on 6 T2D islet preparations [73] showed that, compared to non-diabetic islets, T2D islets showed a reduced insulin content, fewer mature insulin granules, impaired glucose-induced insulin secretion, reduced insulin mRNA expression, increased apoptosis, higher expression of nitrotyrosine (a marker of oxidative stress) and genes involved in redox balance. Remarkably, 24 h exposure of T2D islets to a therapeutically relevant concentration of metformin increased the insulin content, increased the number and density of mature insulin granules, improved glucose-induced insulin release, increased insulin mRNA expression and reduced apoptosis [73]. These effects were associated with decreased oxidative stress, with lower levels of nitrotyrosine and changes in the expression of NADPH oxidase, catalase and GSH peroxidase after metformin exposure [73].

Incretin molecules may also have direct beneficial effects on T2D islets. Islets from 7 T2D and 11 non-diabetic donors were exposed for 48 h to 10 nmol/L exendin-4, a DPP-4-resistant GLP-1 mimetic [70]. Exendin-4 improved glucose-stimulated insulin release from both T2D and non-diabetic islets. In diabetic islets the expression of key beta cell genes was increased, including glucokinase, PDX-1, E2F1 and Cyclin D1. A couple of years later, the same group assessed the direct effect of GLP-1 and GIP alone or in combination on 4 T2D and non-diabetic islets [74]. Islets were exposed to incretins for 45 min (acute exposure: 0.1, 1, 10 or 100 nmol/L) or 2 days (prolonged exposure: 10 nmol/L). Acute exposure (at 1–100 nmol/L) to GLP-1 and, more markedly so, GIP, improved glucose-stimulated insulin release in non-diabetic islets, with no apparent synergistic action. Similar effects were observed with T2D islets acutely treated with 10 nmol/L GLP-1 or 100 nmol/L GIP. Following prolonged exposure, improved insulin secretion was observed with T2D islets cultured with GIP or GLP-1 in combination with GIP. Insulin, PDX-1 and Bcl-2 expression tended to be higher after incretin exposure in both T2D and non-diabetic islets, with the incretin combination showing more robust effects. The mechanisms underlying the beneficial effects of incretins on human beta cells are not fully understood, but it has been reported that GLP-1 receptor agonism induces the endoplasmic reticulum chaperone BiP and the antiapoptotic protein JunB [71].

More recent work [75] investigated the effect of modulating autophagy (that leads to the degradation and recycling of intracellular components) on beta cell function, survival and ultrastructure. Islets from 17 non-diabetic and 9 T2D organ donors were cultured for 1–5 days with 10 ng/mL rapamycin (an autophagy inducer), 5 mM 3-methyladenine (3-MA) or 1.0 nM concanamycin-A (two autophagy blockers), in the presence or not of metabolic (0.5 mM palmitate) or chemical (0.1 ng/mL brefeldin A) endoplasmic reticulum stressors. In non-diabetic islets, glucose-stimulated insulin secretion was reduced by palmitate and brefeldin; rapamycin prevented palmitate- but not brefeldin-mediated cytotoxic damage. Palmitate (and in similar way, brefeldin A) exposure increased beta cell apoptosis in non-diabetic islets, which was prevented by rapamycin and worsened by 3-MA. Both palmitate and brefeldin induced the expression of endoplasmic reticulum stress markers (PERK, CHOP and BiP), which was prevented by rapamycin. In T2D islets, rapamycin improved insulin secretion, reduced beta cell apoptosis and preserved insulin granules, mitochondria and endoplasmic reticulum ultrastructure; this was associated with a significant reduction in PERK, CHOP and BiP gene expression. Altered autophagy is associated with beta cell dysfunction and death [76,77], and hyperactivation of mTORC1 (mechanistic target of rapamycin complex 1, that represses the autophagic pathway) has been found in islets from type 2 diabetic donors [78]. In addition, palmitate exposure may result in de-acidification of lysosomes, which impairs autophagy [79]. All this supports the concept that restoring autophagy can improve human beta cell health.

On the whole, the available evidence indicates that beta cell dysfunction in T2D subjects can be rescued by certain treatments that reduce oxidative and endoplasmic reticulum stress and/or promote autophagy.

## 6. Conclusions

Beta cell failure is crucial to the development and progressive deterioration of T2D, and several excellent reviews have focused on the mechanisms leading to beta cell damage, including the relevance of genetic and environmental factors and the role of the intracellular organelles (such as the endoplasmic reticulum and the mitochondria) and pathways involved [1,2,3,4,5,6,7,17,18,19,20,21,22,23,24,25,26,27,80,81,82] However, T2D should no longer be considered a relentlessly worsening disease, as demonstrated by recent evidence of remission of diabetes in variable proportions of patients after a low-calorie diet and bariatric surgery. Here we have reviewed the data that link the rescue of beta cell function with T2D remission. Available in vivo and ex vivo data, obtained in non-diabetic and T2D diabetic subjects and/or pancreatic islets, point to the possibility of directly improving beta cell health by approaches reducing metabolic stress. The associated changes in islet cell molecular traits could represent targets for intervention strategies to promote T2D remission via actions on the beta cells.

## Figures and Tables

**Figure 1 ijms-23-07435-f001:**
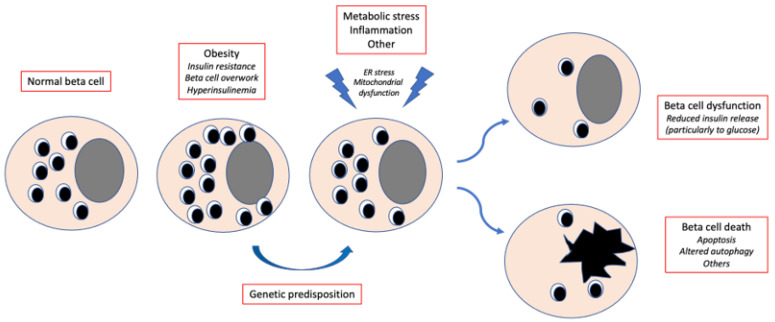
Simplified trajectory of beta cell damage in type 2 diabetes. Obesity and the associated insulin resistance lead to beta cell overwork and the resulting hyperinsulinemia maintains blood glucose levels within the normal range. In case of genetic predisposition, metabolic stress, inflammation and other factors cause alterations in key beta cell organelles, which contributes to beta cell dysfunction and/or death. *ER*, *endoplasmic reticulum.*

**Table 1 ijms-23-07435-t001:** Type 2 diabetes remission: definition and recommendations (adapted from [13,14]).

The term used to describe a sustained metabolic improvement in type 2 diabetes to nearly normal levels should be *remission* of diabetes.
Remission should be defined as a return of HbA1c to <6.5% (<48 mmol/mol) that occurs spontaneously or following an intervention, and that persists for at least 3 months in the absence of usual glucose-lowering pharmacotherapy.
When HbA1c is determined to be an unreliable marker of chronic glycemic control, FPG < 126 mg/dl (<7.0 mmol/L) or estimated A1c < 6.5% calculated from continuous glucose monitoring can be used as alternative criteria.

**Table 2 ijms-23-07435-t002:** Studies showing in vivo recovery of beta cell function.

Ref. in This Article	Number of Participants	Experimental Design	Results
[29]	12 healthy subjects	24 h Intralipid infusion; IVGTT during the infusion and 24 h later	Improved acute insulin release at the IVGTT 24 h after the end of Intralipid infusion
[30]	9 healthy subjects genetically predisposed to T2D	Reduction of plasma NEFA by acipimox; Hyperglycemic clamp 48 h after drug initiation, repeated 2–6 weeks afterwards	Improved first- and second-phase insulin release after reduction of NEFA
[31]	11 T2D subjects	Low calorie diet; 2 square-wave steps of hyperglycaemia after hyperinsulinemic clamp	Improved first-phase insulin release
[32]	382 newly diagnosed T2D patients	Intensive therapy with insulin or oral hypoglycaemic agents	Improved beta cell function after intensive intervention, with more sustained results in the insulin group; Diabetes remission rate at 1 year significantly higher in the insulin group
[33]	12 morbidly obese T2D subjects	Bariatric surgery (distal Roux-en-Y reconstruction); Mixed meal test	Improved insulin secretion
[34]	13 morbidly obese non-diabetic women	Bariatric surgery (distal Roux-en-Y reconstruction); Arginine-infusion at different glucose levels	Improved beta cell function
[35]	10 obese T2D subjects	Bariatric surgery (distal Roux-en-Y reconstruction); IVGTT and mixed meal test	Improved first-phase insulin secretion

T2D, type 2 diabetes; IVGTT, intravenous glucose tolerance test; NEFA, non-esterified fatty acids.

**Table 3 ijms-23-07435-t003:** Studies showing ex vivo recovery of beta cell function.

Ref. in This Article	Number of Samples	Experimental Design	Results
[28]	26 islet preparations from non-diabetic subjects	2-day incubation with lipo-glucotoxic stressors followed by 4 days of washout; Insulin release in response to 3.3 and 16.7 mmol/L glucose during static incubation	Recovery of *beta cell* function after washout of some of the metabolic stressful conditions, associated with specific transcriptomic changes
[67]	7 islet preparations from non-diabetic subjects	48 h incubation at 5.5 or 16.7 mmol/L glucose followed by 48 h washout; Insulin release in response to 3.3 and 16.7 mmol/L glucose during perifusion	Recovery of glucose-stimulated insulin secretion after washout
[68]	4 islet preparations from non-diabetic subjects	2 preparations incubated at 5.5 mmol/L glucose and 2 at 33 mmol/L glucose for 4–9 days; 1 preparation of the latter incubated for 6 days at 33 mmol/L glucose and then for 3 days in normal glucose; Insulin release in response to 3.3 and 16.7 mmol/L glucose during static incubation	Recovery of insulin mRNA expression and insulin secretion after washout
[69]	6 islet preparations from T2D donors	24 h incubation with metformin; Insulin release in response to 3.3 and 16.7 mmol/L glucose during static incubation or perifusion	Improved glucose-stimulated insulin release; Beneficial action on the expression of genes involved in redox balance: Ameliorated beta cell ultrastructure
[70]	18 islet preparations (11 non-diabetic and 7 T2D donors)	Exposure to exendin-4 for 48 h; Insulin release in response to 3.3 and 16.7 mmol/L glucose during static incubation	Improved glucose-stimulated insulin release; Beneficial actions on expression of genes involved in beta cell function and identity
[71]	26 islet preparations (17 non-diabetic and 9 T2D donors)	Exposure to autophagy inducers (rapamycin) or blockers (3-methyladenince, concanamycin-A) for 2–5 days, in the presence or not of palmitate or brefeldin A; Insulin release in response to 3.3 and 16.7 mmol/L glucose during static incubation	Rapamycin-exposed diabetic islets showed improved insulin secretion, reduced apoptosis and better ultrastructure

T2D, type 2 diabetes.

## Data Availability

Not applicable.

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
