# Peer review of "The Role of Beta Cell Recovery in Type 2 Diabetes Remission"

_ijms, 2022, doi:10.3390/ijms23137435_

Round 1

Reviewer 1 Report

This is a timely review on possible mechanisms of beta-cell remission in T2D motivated by a recent observational study in >150,000 patients with T2D reporting remission of T2D in 5% of cases (Ref15).

Authors summarise the results of several studies reporting recovery of beta cell function and survival in vitro and in vivo. The message is clear and review well written, I only have some suggestions for a more comprehensive look at previous literature.

In their simplified Fig.1, they omit the usually occurring compensation phase before beta-cell failure establishes; with hyperinsulinemia, in which most obese individuals (without genetic predisposition) remain throughout life, this should be included in the figure.

This whole part of compensation also needs to be included as it is an important path linked to the mechanism of beta-cell failure: the mTORC pathway plays a prominent role here, as on the long term, its chronic hyperactivation (by over-nutrition/glucose/FFA) leads to beta-cell collapse, while beta-cell remission can be induced by blocking the mTORC1 path, nicely illustrated recently in mice during HFD feeding (Am J Pathol. 2022 Jan;192(1):130-145) or in T2D islets long before (Diabetologia. 2017 Apr;60(4):668-678 & Cell Metab. 2018 Feb 6;27(2):314-331). These studies provide a mechanistical explanation of earlier data from the authors showing similar remission of T2D islets by metformin (Ref [71]) which stimulates AMPK, subsequently inhibiting mTORC1 and promoting autophagy-the supportive effect of mTORC inhibition (and authophagy stimulation) has been confirmed by the authors later (Ref 75)

It is well-appreciated that the authors find lipotoxicity & glucotoxicity as major causes for the deterioration and apoptosis of the beta-cell and therefore should not ignore earlier findings in human islets in vitro by the Donath lab (Diabetes. 2001 Jan;50(1):69-76. Diabetes. 2001 Aug;50(8):1683-90), among their own confirmations of this concept.

Page 5, line 165, change XXX to (5.5 mmol/l)

Author Response

The authors are grateful to the Reviewer for her/his insightful comments. All the suggestions have been taken into consideration, as detailed below.

1) Figure 1 has been amended as suggested, to show the compensation phase that, in case of genetic predisposition, can be followed by the decline of beta cell health.

2) Although the description of the mechanisms leading to beta cell damage is beyond the purpose of this manuscript, the points raised by the Reviewer are well taken. Therefore, we have added a sentence (page 5) and a reference (# 78) to better explain the mTORC1 issue. Similarly, ref. 4 has been added to support the first sentence of the manuscript and document the inflammatory damage induced by metabolic stress in T2D islets.

3) The typo at page 5 has been corrected.

Reviewer 2 Report

Suleiman et al. provide a concise overview of the literature on beta cell recovery after onset of T2D.  Their review nicely summarizes studies that show progression of T2D is not irreversible and that specific changes in diet and/or pharmacological intervention could rescue islet function.  This review could benefit by highlighting the cell biology that underlies the normal function of beta cells and discussing how nutrient excess is known to stress organelles and dysregulate their ability to sense glucose and promote insulin secretion.  Moreover, the role of exercise in reversing diabetes is an increasingly important area of research, and the authors should make an effort to underscore this important point.  

Major comments:

The graphics and text in Fig. 1 are pixelated.  The authors should use a higher dpi image, so it is legible.

Lines 64-69:  In the graphic in Fig. 1, the authors highlight mitochondrial dysfunction as a factor in the progression of T2D.  Given the role of mitochondria in glucose and pyruvate metabolism, as well as the fundamental role of mitochondria in ATP production, it is important to discuss and cite the literature that links mitochondrial impairment with the onset and progression of T2D.  It is worth noting, for example, studies showing that lipotoxicity/glucolipotoxicity can induce mitochondrial fragmentation and depolarize the mitochondrial membrane potential, which has downstream effects on insulin secretion (lines 103-105).

Lines 210-215:  Is all of this data associated with citation 71 (line 206)?  If not, please provide relevant citations.  Metformin lowered catalase expression in the beta cell (line 215)?  It is not clear how this could be associated with lower levels of oxidative stress, since catalase is an antioxidant enzyme.  Furthermore, catalase expression is already very low in healthy mammalian beta cells, so this observation seems questionable.  Authors should double-check the literature and ascertain whether these are indeed the effects of metformin on beta cell physiology, specifically.

Lines 234-250:  Given the authors’ emphasis of the role of lipotoxicity in beta cell dysfunction, it is worth noting that studies have shown that exposure to elevated levels of specific fatty acids, such as palmitate, results in de-acidification of lysosomes, which impairs autophagy.  This provides a mechanistic link to the consumption of fatty acids and the impaired turnover of mitochondria, ER, and other cellular components that, together, impact nutrient sensing and insulin secretion.

Given the connections between fasting/caloric restriction and exercise (particularly aerobic exercise), the authors should discuss studies that investigate exercise as a healthy approach for rescuing insulin secretion and sensitizing peripheral tissues to circulating insulin.

Minor comments:

Line 78:  “after at 6 h” should be “at 6 h”

Line 148:  “but no all patients” should be “but not all patients”

Author Response

We thank the Reviewer for her/his suggestions, that we have taken into careful consideration in the revised version of the manuscript. In particular:

1) Figure 1 has changed in content and format, and we hope now it is better readable

2) We agree with the reviewer that the role of mitochondria is of outmost importance in affecting beta cell health and function. However, a detailed description of the mitochondrial mechanisms that lead to beta cell dysfunction is beyond the purposes of this manuscript (focused on data showing human beta cell function rescue). Nevertheless, given the relevance of the point, we have added a sentence at the beginning of the discussion and a few additional references (# 81 and 82) on this issue.

3) The point related to lysosome de-acidification by palmitate is very interesting, and we have briefly dealt with it (end of page 8) and added ref. # 79.

4) We agree that physical exercise is a very important approach, that usually, at least in humans, has been used in combination with nutritional strategies. This has been clarified  at lines 124-125.

5) The mistakes reported in Minor Comments have been corrected.